# Diagnostic accuracy of CT angiography and CT perfusion imaging for detecting distal medium vessel occlusions: Protocol for a systematic review and meta-analysis

João André Sousa[1] º*, Anton Sondermann[2]º, Sara Bernardo-Castro[1,3], Ricardo Varela[4], Helena Donato[5], João Sargento-Freitas[1,3]

1 Neurology Department, Centro Hospitalar e Universitário de Coimbra, Coimbra, Portugal, 2 Christian-Albrechts-Universität zu Kiel, Kiel, Germany, 3 Faculdade de Medicina da Universidade de Coimbra, Coimbra, Portugal, 4 Neurology Department, Centro Hospitalar e Universitário de Santo António, Porto, Portugal, 5 Documentation and Scientific Information Service, Centro Hospitalar e Universitário de Coimbra, Coimbra, Portugal

º These authors contributed equally to this work.

* sousajoaoandre@gmail.com

**Data Availability Statement:** No datasets were generated or analysed during the current study. All

## Abstract

### Background

Distal medium vessel occlusions (DMVOs) represent 25–40% of all acute ischemic strokes (AIS). DMVO clinical syndromes are heterogenous, but as eloquent brain regions are frequently involved, they are often disabling. Since current intravenous fibrinolytic therapies may fail to recanalize up to two-thirds of DMVOs, endovascular treatment is progressively being considered in this setting. Nevertheless, the optimal imaging method for diagnosis remains to be defined. Stroke centers that use computed tomography as a routine stroke imaging approach rely on either isolated computed tomography angiography (CTA) or combined perfusion (CTP) studies. Despite a simplified non-CTP-dependent approach seeming reasonable for large vessel occlusion AIS diagnosis, CTP may still hold advantages for DMVOs workup. Therefore, this systematic review aims to compare the diagnostic performance of CTA and CTP in detecting DMVOs.

### Methods

We will perform a systematic search in PubMed, EMBASE, Web of Science Core Collection, and Cochrane Central Register of Controlled Trials. In addition, grey literature and ClinicalTrials.gov will be scanned. We will include any type of study that presents data on the diagnostic accuracy of CTA and/or CTP for detecting DMVOs. Two authors will independently review retrieved studies, and any discrepancies will be resolved by consensus or with a third reviewer. Reviewers will extract the data and assess the risk of bias in the selected studies. Data will be combined in a quantitative meta-analysis following the guidelines provided by the Cochrane Handbook for Systematic Reviews of Interventions. We will assess

relevant data from this study will be made available upon study completion.

**Funding:** The authors received no specific funding for this work.

**Competing interests:** The authors have declared that no competing interests exist.

cumulative evidence using the Grading of Recommendations, Assessment, Development and Evaluation (GRADE) approach.

## Discussion

This will be the first systematic review and meta-analysis that compares two different imaging approaches for detecting DMVOs. This study may help to define optimal acute ischemic stroke imaging work-up.

## Trial registration

**PROSPERO registration:** CRD42022344006.

## Introduction

Distal medium vessel occlusions (DMVOs) are mid-sized artery occlusions with lumen diameters between 0.75 and 2.0 mm, representing 25–40% of all acute ischemic strokes [1–3]. This category includes M2 to M4 segments of the medial cerebral artery (MCA), A2 to A5 of the anterior cerebral artery (ACA), and P2 to P5 of the posterior cerebral artery (PCA) [1]. DMVOs are not benign. For instance, M2 occlusions can have a high National Institute of Health Stroke Score (NIHSS) (>8–10) and consequently high associated mortality ranging from 9.5% (HERMES sub-study) [4] to 41% (STOPStroke Study) [5]. As intravenous thrombolysis may fail to recanalize more than half of DMVOs [2], endovascular therapy (EVT) is progressively considered in this setting, as up to M4-MCA segments may be safely approached [1]. ESCAPE-MeVO trial (NCT05151172) and DISTAL trial (NCT05029414) are underway in recruiting patients to evaluate if EVT is superior to best medical management in DMVOs. As current acute stroke imaging protocols were developed and tested to detect large vessel occlusions, optimal imaging approaches to detect DMVOs are yet to be refined. There is a trend towards simple non-contrast computed tomography (NCCT) and angiography (CTA) to select patients for EVT even in the late-window [6]. Still, DMVOs are challenging to detect on CTA as up to one-third of DMVOs can be missed [7, 8]. CTP may hold improved accuracy for DMVOs detection, but this imaging modality is not accessible in many stroke centers and may be time-consuming [9, 10].

We aim to compare diagnostic accuracy of CTA and CTP regarding detection of DMVOs by performing a systematic review and meta-analysis.

## Material and methods

This review of scientific literature will follow the methodological guidelines published in the PRISMA statement for Diagnostic Test Accuracy (DTA) (PROSPERO protocol ID CRD42022344006) [11].

### Eligibility criteria

We will identify randomized controlled trials (RCTs), cohort studies (prospective or retrospective), cross-sectional studies, and case-controlled studies that quantify the diagnostic accuracy of CTA and/or CTP in detecting DMVOs. Studies fulfilling the eligibility criteria shown in Table 1 will be selected for further review. Table 2 presents our PICO description. If the same study is reported in different articles, the one with the largest sample size or presenting data

**Table 1. Inclusion/exclusion criteria.**

| Inclusion criteria |
| --- |
| Studies on living humans |
| Original data |
| Acute ischemic stroke |
| M2, M3, M4, A2, A3, A4, A5, P2, P3, P4, P5, PICA, AICA or SCA artery occlusions |
| Studies using head CT angiography and/or perfusion and reporting diagnostic values of such imaging methods |
| All study designs (prospective and retrospective) |
| From inception to 31th December 2022 |
| **Exclusion criteria** |
| ICA, M1, A1, P1, VA and BA artery occlusions |
| Lacunar strokes |
| Case reports |

more suitable for our specific aim will be selected. We will include studies from inception to the 31st of December 2022. No language restriction will be applied. We will resort to a professional translator if needed.

## Information sources

We will identify published literature in the following databases: PubMed, EMBASE, Web of Science Core Collection, and Cochrane Central Register of Controlled Trials. We will also consult ClinicalTrials.gov to check published or unpublished trials. If any relevant unpublished trial is found, the corresponding author listed will be contacted to grant access to the required information. If no response is given or the author decides not to share the data, this will be listed as the reason for excluding such a trial. Grey literature will also be queried to include all possible articles on the subject. No pre-prints will be included.

## Search strategy

Detailed search strategies designed for each database are presented in Table 3. Different combinations of the following terms will be used: "acute ischemic stroke", "distal vessel occlusion", "medium vessel occlusion", "distal medium vessel occlusion", "computed tomography angiography", "computed tomography perfusion", "perfusion imaging" and database-specific subject headings (e.g., MeSH terms). Regarding grey literature, we will scan the references of selected studies and reviews on the subject as well as conference papers.

## Data management

All studies obtained after the literature search will be imported to Rayyan QCRI Software [12], where duplicates will be managed and erased, and titles/abstracts of all records will be scanned.

**Table 2. PICO description.**

| Abbreviation | PICO | Elements |
| --- | --- | --- |
| P | Population | Acute ischemic stroke caused by a distal, medium vessel occlusion |
| I | Intervention/Exposure | Computed tomography perfusion |
| C | Comparator | Computed tomography angiography |
| O | Outcome | Detection of the occlusion |

**Table 3. Search strategy.**

| PubMed | |
| --- | --- |
| **Query** | **Search** |
| #1 | "Ischemic Stroke"[Mesh] OR "Arterial Occlusive Diseases"[Mesh] OR Stroke OR "Acute stroke" OR "Acute Ischemic Stroke" OR Apoplexy OR "Cerebral Apoplexy" OR "Cerebrovascular Accident" OR "Brain Vascular Accident" OR CVA |
| #2 | "Distal Medium Vessel Occlusion" OR "Medium Vessel Occlusion" OR MeVO OR DMVO OR "Distal Artery Occlusion" OR M2 OR M3 OR M4 OR A2 OR A3 OR A4 OR A5 OR P2 OR P3 OR P4 OR P5 OR PICA OR "Posterior Inferior Cerebellar Artery" OR AICA OR "Anterior Inferior Cerebellar Artery" OR SCA OR "Superior Cerebellar Artery" |
| #3 | "Computed Tomography Angiography"[Mesh] OR "Perfusion Imaging"[Mesh] OR "Tomography, X-Ray Computed"[Mesh] OR "Perfusion Computed Tomography" OR "Computed Tomography" OR CT OR CTA OR CTP |
| #4 | Search #1 AND #2 AND #3 |
| **EMBASE** | |
| #1 | 'cerebrovascular accident'/exp OR 'brain ischemia'/exp OR 'cerebrovascular accident' OR 'stroke patient' OR 'brain ischemia' OR 'stroke' OR 'acute ischemic stroke' OR 'ischemic stroke' OR 'apoplexy' OR 'cerebral apoplexy' OR 'brain apoplexy' |
| #2 | 'distal medium vessel occlusion' OR 'medium vessel occlusion' OR 'MeVO' OR 'DMVO' OR 'm2' OR 'm3' OR 'm4' OR 'a2' OR 'a3' OR 'a4' OR 'a5' OR 'p2' OR 'p3' OR 'p4' OR 'P5' OR 'posterior inferior cerebellar artery'/exp OR 'PICA' OR 'anterior inferior cerebellar artery'/exp OR 'AICA' OR 'superior cerebellar artery'/exp OR 'SCA' |
| #3 | 'computed tomographic angiography'/exp OR 'multidetector computed tomography'/exp OR 'CTA' OR 'CT angiography' OR 'computer assisted tomography'/exp OR 'computed tomography' OR 'ct' OR 'perfusion computed tomography'/exp OR 'perfusion computer assisted tomography'/exp OR 'perfusion computer tomography'/exp OR 'CT perfusion' OR 'CTP' |
| #5 | #1 AND #2 AND #3 |
| **Cochrane** | |
| #1 | MeSH descriptor: [Stroke] explode all trees |
| #2 | Stroke OR "Acute stroke" OR "Acute Ischemic Stroke" OR Apoplexy OR "Cerebral Apoplexy" OR "Cerebrovascular Accident" OR "Brain Vascular Accident" OR CVA |
| #3 | MeSH descriptor: [Computed Tomography Angiography] explode all trees |
| #4 | MeSH descriptor: [Perfusion Imaging] explode all trees |
| #5 | MeSH descriptor: [Tomography Scanners, X-Ray Computed] explode all trees |
| #6 | MeSH descriptor: [Neuroimaging] explode all trees |
| #7 | MeSH descriptor: [Diagnostic Imaging] explode all trees |
| #8 | "Computed Tomography" OR "Perfusion Computed Tomography" OR "Computed Tomography Angiography" OR "CT" OR "CTP" OR "CTA" |
| #9 | "Distal Medium Vessel Occlusion" OR "Medium Vessel Occlusion" OR MeVO OR DMVO OR "Distal Artery Occlusion" OR M2 OR M3 OR M4 OR A2 OR A3 OR A4 OR A5 OR P2 OR P3 OR P4 OR P5 OR PICA OR "Posterior Inferior Cerebellar Artery" OR AICA OR "Anterior Inferior Cerebellar Artery" OR SCA OR "Superior Cerebellar Artery" |
| #10 | (#1 OR #2) AND (#3 OR #4 OR #5 OR #6 OR #7 OR #8) AND #9 |
| **Web of Science** | |
| #1 | TS = ('cerebrovascular accident' OR 'brain ischemia' OR 'stroke patient' OR 'ischemic stroke' OR 'ischemic stroke' OR 'stroke' OR 'apoplexy' OR 'cerebral apoplexy' OR 'brain apoplexy') AND TS = ('computed tomographic angiography' OR 'multidetector computed tomography' OR 'CTA' OR 'CT angiography' OR 'computer assisted tomography' OR 'computed tomography' OR 'ct' OR 'perfusion computed tomography' OR 'perfusion computer assisted tomography' OR 'perfusion computer tomography' OR 'CT perfusion' OR 'CTP') AND TS = ('distal medium vessel occlusion' OR 'medium vessel occlusion' OR 'm2' OR 'm3' OR 'm4' OR 'a2' OR 'a3' OR 'a4' OR 'a5' OR 'p2' OR 'p3' OR 'p4' OR 'P5' OR 'posterior inferior cerebellar artery' OR 'PICA' OR 'anterior inferior cerebellar artery' OR 'AICA' OR 'superior cerebellar artery' OR 'SCA') |

## Selection process

Two independent reviewers will conduct the selection phase of the studies during the month of March 2023. Disagreement will be resolved by consensus or by a third party if necessary. All records identified in the search stage will be screened by title/abstract. Studies not matching the criteria and duplicate studies will be excluded. The remaining studies will be thoroughly read. Reasons for the exclusion of full-text records will be recorded. Rayyan QCRI Software will be used to perform these steps, and Mendeley Software will be used to format the references. A flow chart similar to Fig 1 will be presented to detail the study selection process for this review.

## Data collection process

Two reviewers will independently extract information from the articles to ensure all necessary details are obtained from the selected studies and minimize the risk of bias. Disagreement will be resolved by consensus or by third-party arbitrage. The data extracted will be reviewed and validated by a third reviewer. This will occur after the selection process.

## Data items

For included studies, we will collect the index test, sample size, and either individual data or summary estimates of the number of true positive, true negative, false positive, false negative, sensitivity, specificity, positive predictive value, and negative predictive value of CTA and/or CTP. We will also retrieve information regarding the study design, date, authors, and predominant location of DMVOs.

## Risk of bias (quality) assessment

The methodological quality of all studies included in the systematic review will be assessed independently by two reviewers. We will use QUADAS-2 [13] to examine the risk of bias and applicability of the collected primary diagnostic accuracy studies concerning four separate domains: (a) patient selection, (b) index test (diagnostic technique investigated in the study), (c) reference standard (the ground truth technique used as reference), and (d) flow and timing in the study. For each QUADAS2 domain, any concern regarding bias and applicability will be scored as "high," "low," or "unclear", based on the information presented by the authors in each publication.

## Data synthesis and sensitivity analysis

This systematic review will include a quantitative meta-analysis. The statistical analysis will follow the guidelines from the Cochrane Handbook for Systematic Reviews of Interventions [14] and will be performed using R software v.4.2.1 and RStudio. We will collect the data mentioned above from each study and compute the sensitivity, specificity, positive likelihood ratio, and negative likelihood ratio. We will use a bivariate model to produce summary receiver operating curves, pooled sensitivity and specificity, and 95% confidence regions around the summary operating point. We will test the consistency and heterogeneity of the studies with the Higgins $I^2$ statistic. Following the direction given by Higgins et al., 2003 we will consider 25% as low heterogeneity, between 25 and 50% moderate heterogeneity, and 75% as high heterogeneity. If the $I^2$ value is ≤50% (low to moderate heterogeneity), we will use the fixed effect model for data synthesis; if it is greater than 50%, we will use the random effects model. If the heterogeneity values are over 75%, we will search for the possible sources of this high heterogeneity, including reviewing the methodological processes of the selected studies and searching

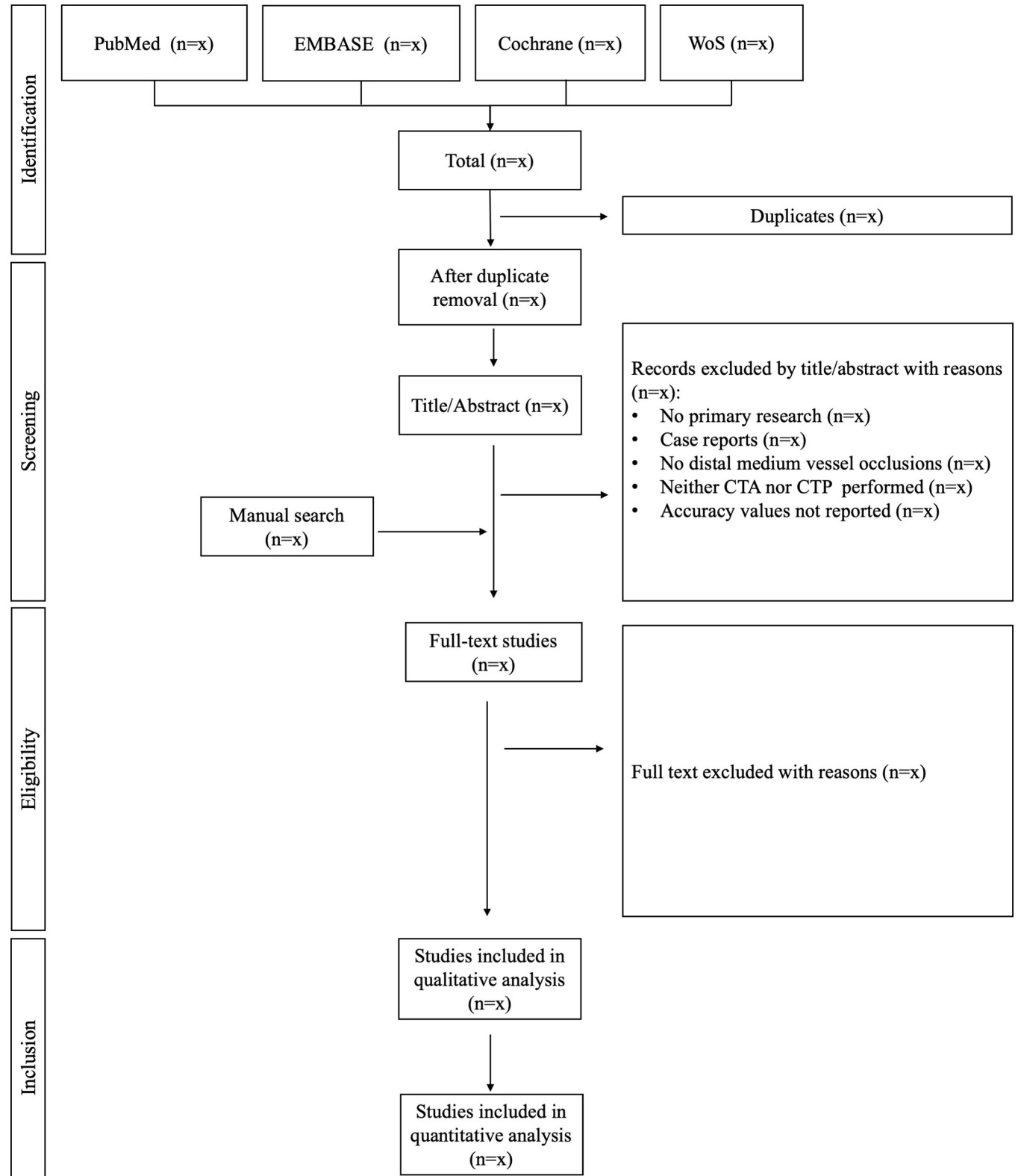

**Fig 1. Flowchart diagram presenting the selection process for the studies.**

for outliers or influential cases that may distort the analysis results. Any possible outlier or significant case, as well as studies presenting poor methodological quality and/or a high or critical risk of bias, will be excluded in a further sensitivity analysis.

### Subgroup analysis

We will include a subgroup analysis according to the technique used in CTA (single phase or multiphase) as different accuracy values have been suggested between both algorithms [15, 16].

### Meta-bias(es)

To assess publication bias, we will perform a funnel plot following the recommendation of the Cochrane Handbook for Systematic Reviews of Interventions [14] and a complemental Egger's test to quantify the funnel plot's asymmetry if possible ($k \geq 10$ studies).

### Confidence in cumulative evidence

The strength of the body evidence will be assessed using the Grading Recommendations Assessment, Development, and Evaluation (GRADE) [17].

### Patient and public involvement

No participant recruitment will occur as this is a secondary study based on previously published literature.

### Ethics and dissemination

This work consists of a secondary study based on public and published data; therefore, ethical approval is unnecessary. The result obtained from this work will be published in a peer-reviewed journal and disseminated at relevant conferences. If any amendments are needed due to deviations from this protocol in the execution of the study, these amendments will be recorded and noted in the publication. Aggregate data from this study will be available upon reasonable request from the corresponding author.

## Discussion

This will be the first systematic review and meta-analysis to address the identification of the optimal imaging modality to detect DMVOs.

This is an ever more treatable condition either by thrombolysis and/or thrombectomy. Six years after the conquest of large vessel occlusion acute ischemic stroke by pivotal endovascular treatment trials, the focus of research is on new targets, including DMVOs. At the upstream of the treatment of this condition, there is a need to diagnose it accurately, but the ideal imaging approach remains to be determined.

This study has limitations as it is a secondary study dependent on the published data. Moreover, a key element in diagnostic accuracy studies is the reference standard or the test used to define the condition. Digital subtraction cerebral angiography (DSA) is considered the "gold standard" for evaluating intracranial and extracranial vessels of the brain. However, DSA is generally reserved for cases where a vessel occlusion is detected in another exam and endovascular treatment is expected. Therefore, we do not expect to find studies where CTA and/or CTP are compared to DSA but rather CTA vs. CTP, knowing that the sensitivity of both exams can be suboptimal. We expect to have a manuscript ready to be submitted at the beginning of May 2023.

## Supporting information

**S1 Checklist. PRISMA-P (Preferred Reporting Items for Systematic review and Meta-Analysis Protocols) 2015 checklist: Recommended items to address in a systematic review protocol\*.**
(DOC)

## Author Contributions

**Conceptualization:** João André Sousa, Ricardo Varela, João Sargento-Freitas.

**Investigation:** João André Sousa, Anton Sondermann, Sara Bernardo-Castro.

**Methodology:** João André Sousa, Anton Sondermann, Sara Bernardo-Castro, Ricardo Varela, Helena Donato.

**Supervision:** João Sargento-Freitas.

**Writing – original draft:** João André Sousa, Anton Sondermann.

**Writing – review & editing:** Sara Bernardo-Castro, Ricardo Varela, Helena Donato, João Sargento-Freitas.

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
