## [Decision Letter · Decision Letter 0]

17 Feb 2023

PONE-D-22-25227Diagnostic accuracy of CT angiography and CT perfusion imaging for detecting distal medium vessel occlusions: protocol for a systematic review and meta-analysisPLOS ONE

Dear Dr. Sousa,

Thank you for submitting your manuscript to PLOS ONE. After careful consideration, we feel that it has merit but does not fully meet PLOS ONE’s publication criteria as it currently stands. Therefore, we invite you to submit a revised version of the manuscript that addresses the points raised during the review process.

Please address all reviewer 2's comments thoroughly. 

We look forward to receiving your revised manuscript.

Kind regards,

Stephan Meckel, MD, PhD

Academic Editor

PLOS ONE

Reviewers' comments:

Reviewer's Responses to Questions

**Comments to the Author**

1. Does the manuscript provide a valid rationale for the proposed study, with clearly identified and justified research questions?

Reviewer #1: Yes

Reviewer #2: Yes

2. Is the protocol technically sound and planned in a manner that will lead to a meaningful outcome and allow testing the stated hypotheses?

Reviewer #1: Yes

Reviewer #2: Yes

3. Is the methodology feasible and described in sufficient detail to allow the work to be replicable?

Reviewer #1: Yes

Reviewer #2: Yes

4. Have the authors described where all data underlying the findings will be made available when the study is complete?

Reviewer #1: Yes

Reviewer #2: No

5. Is the manuscript presented in an intelligible fashion and written in standard English?

Reviewer #1: Yes

Reviewer #2: Yes

6. Review Comments to the Author

You may also provide optional suggestions and comments to authors that they might find helpful in planning their study.

Reviewer #1: The authors provide an adequate protocol for an SR-MA on middle and distal occlusions. Looking forward to the study.

Reviewer #2: This paper outlines a protocol for a systematic review testing the diagnostic accuracy of CTP vs. CTA for detection of distal vessel occlusions. The protocol is, apart from a few minor errors, for the most part well written and straightforward. I have only a few comments:

- Please include the ongoing ESCAPE-MeVO trial (in addition to DISTAL) in the introduction

- In the Methods, the studies included go to August 31st, 2022, but in Table 1, July 2022 is written. Please correct this discrepancy

- Please include a statement regarding data availability upon completion of the study

- Please capitalize all months, re-read the manuscript for minor typographical errors (e.g., Page 6, under Data synthesis and sensitivity analysis, "we will collect the data mentioned above items.....". The word "items" should be removed)

- I think it would be interesting to perform an analysis of mCTA vs. CTP, as this has been shown to be at least equivalent, if not better, at DMVO detection in several single-center studies. Since mCTA can be generated with a minimal increase in time, no increase in radiation dose, etc., it seems like a more interesting comparison than to spCTA or CTA in general.

7. PLOS authors have the option to publish the peer review history of their article (what does this mean?). If published, this will include your full peer review and any attached files.

Reviewer #1: No

Reviewer #2: No

---

## [Author Response · Author response to Decision Letter 0]

4 Mar 2023

We are grateful for the thorough and valuable manuscript review that led to a considerable improvement of the article. Below we detail the answers to the provided remarks:

Comment: “-Please include the ongoing ESCAPE-MeVO trial (in addition to DISTAL) in the introduction”

Answer: We have included the ESCAPE-MeVO trial in the Introduction section: “ESCAPE-MeVO trial (NCT05151172) and DISTAL trial (NCT05029414) are underway in recruiting patients to evaluate if EVT is superior to the best medical management in DMVOs.”

Comment: “In the Methods, the studies included go to August 31st, 2022, but in Table 1, July 2022 is written. Please correct this discrepancy”

Answer: We have corrected and updated our target months in the manuscript. Included articles will be from inception to the 31st of December 2022, selection phase of the studies will occur during the current month of March. We expect the manuscript to be ready for submission at the beginning of May 2023.

Comment: ”Please include a statement regarding data availability upon completion of the study”

Answer: The following sentence has been included in Material and Methods section under the Ethics and dissemination subtopic: “Aggregate data from this study will be available upon reasonable request from the corresponding author”

Comment “-Please capitalize all months, re-read the manuscript for minor typographical errors (e.g., Page 6, under Data synthesis and sensitivity analysis, "we will collect the data mentioned above items.....". The word "items" should be removed)”

Answer: We have capitalized all months throughout the manuscript. Additionally, we have thoroughly re-read the manuscript and corrected minor typographical errors, such as removing the word "items" from the sentence you mentioned on Page 6 under Data synthesis and sensitivity analysis.

Comment “I think it would be interesting to perform an analysis of mCTA vs. CTP, as this has been shown to be at least equivalent, if not better, at DMVO detection in several single-center studies. Since mCTA can be generated with a minimal increase in time, no increase in radiation dose, etc., it seems like a more interesting comparison than to spCTA or CTA in general.”

Answer: We absolutely agree with the reviewer. In fact, we were planning to do that subgroup analysis although it was not written in the protocol. We included the subtopic “Subgroup analysis” in the Material and Methods section where we added the following paragraph: “We will include a subgroup analysis according to the technique used in CTA (single phase or multiphase) as different accuracy values have been suggested between both algorithms[15,16]”

15 McDonough R V, Qiu W, Ospel JM, et al. Multiphase CTA-derived tissue maps aid in detection of medium vessel occlusions. Neuroradiology 2022;64:887–96. doi:10.1007/s00234-021-02830-8

16 Ospel JM, Bala F, McDonough R V, et al. Interrater Agreement and Detection Accuracy for Medium-Vessel Occlusions Using Single-Phase and Multiphase CT Angiography. AJNR Am J Neuroradiol 2022;43:93–7. doi:10.3174/ajnr.A7361

---

## [Decision Letter · Decision Letter 1]

27 Mar 2023

Diagnostic accuracy of CT angiography and CT perfusion imaging for detecting distal medium vessel occlusions: protocol for a systematic review and meta-analysis

PONE-D-22-25227R1

Dear Dr. Sousa,

We’re pleased to inform you that your manuscript has been judged scientifically suitable for publication and will be formally accepted for publication once it meets all outstanding technical requirements.

Kind regards,

Stephan Meckel, MD, PhD

Academic Editor

PLOS ONE

Additional Editor Comments (optional):

Reviewers' comments:

Reviewer's Responses to Questions

**Comments to the Author**

1. Does the manuscript provide a valid rationale for the proposed study, with clearly identified and justified research questions?

Reviewer #2: Yes

2. Is the protocol technically sound and planned in a manner that will lead to a meaningful outcome and allow testing the stated hypotheses?

Reviewer #2: Yes

3. Is the methodology feasible and described in sufficient detail to allow the work to be replicable?

Reviewer #2: Yes

4. Have the authors described where all data underlying the findings will be made available when the study is complete?

Reviewer #2: Yes

5. Is the manuscript presented in an intelligible fashion and written in standard English?

Reviewer #2: Yes

6. Review Comments to the Author

You may also provide optional suggestions and comments to authors that they might find helpful in planning their study.

Reviewer #2: Thank you for your edits, all comments have been satisfactorily addressed, I have no further comments to the authors

7. PLOS authors have the option to publish the peer review history of their article (what does this mean?). If published, this will include your full peer review and any attached files.

Reviewer #2: No

---

## [Editor Report · Acceptance letter]

31 Mar 2023

PONE-D-22-25227R1 

Diagnostic accuracy of CT angiography and CT perfusion imaging for detecting distal medium vessel occlusions: protocol for a systematic review and meta-analysis 

Dear Dr. Sousa:

I'm pleased to inform you that your manuscript has been deemed suitable for publication in PLOS ONE. Congratulations! Your manuscript is now with our production department. 

Kind regards, 

on behalf of

Prof. Dr. Stephan Meckel 

Academic Editor

PLOS ONE